

# Genome-wide identification and analysis of ascorbate peroxidase (APX) gene family in hemp (*Cannabis sativa* L.) under various abiotic stresses

Zixiao Liang[1], Hongguo Xu[1,2], Hongying Qi[1,2], Yiying Fei[1] and Jiaying Cui[1]

[1] College of Life Sciences and Agroforestry, Qiqihar University, Qiqihar City, Heilongjiang Province, China
[2] Key Laboratory of Resistance Genetic Engineering and Cold Biodiversity Conservation, Qiqihar University, Qiqihar City, Heilongjiang Province, China

Corresponding author
Hongguo Xu, 01303@qqhru.edu.cn

## ABSTRACT

Ascorbate peroxidase (APX) plays a critical role in molecular mechanisms such as plant development and defense against abiotic stresses. As an important economic crop, hemp (*Cannabis sativa* L.) is vulnerable to adverse environmental conditions, such as drought, cold, salt, and oxidative stress, which lead to a decline in yield and quality. Although *APX* genes have been characterized in a variety of plants, members of the *APX* gene family in hemp have not been completely identified. In this study, we (1) identified eight members of the *CsAPX* gene family in hemp and mapped their locations on the chromosomes using bioinformatics analysis; (2) examined the physicochemical characteristics of the proteins encoded by these *CsAPX* gene family members; (3) investigated their intraspecific collinearity, gene structure, conserved domains, conserved motifs, and cis-acting elements; (4) constructed a phylogenetic tree and analyzed interspecific collinearity; and (5) ascertained expression differences in leaf tissue subjected to cold, drought, salt, and oxidative stresses using quantitative real-time-PCR (qRT-PCR). Under all four stresses, *CsAPX6*, *CsAPX7*, and *CsAPX8* consistently exhibited significant upregulation, whereas *CsAPX2* displayed notably higher expression levels under drought stress than under the other stresses. Taken together, the results of this study provide basic genomic information on the expression of the APX gene family and pave the way for studying the role of APX genes in abiotic stress.

# INTRODUCTION

Plants are exposed to a variety of stresses throughout their life cycle, including abiotic factors, such as cold, oxidation, salt, and drought, as well as biotic factors, such as diseases and pests (*Bigot et al., 2018*; *Bhat et al., 2019*). Adverse environments trigger the production of reactive oxygen species (ROS) in plants. To cope with these stresses, plants disrupt the balance of ROS, which seriously harms their normal growth and development. ROS are extremely important to the physiological processes in plants by serving as essential

signaling molecules and potential agents of cellular damage (*Gilroy et al., 2016*). Hydrogen peroxide ($H_2O_2$) and superoxide radicals ($O^{2-}$) are signaling molecules that participate in the regulation of various signal networks in plants (*Zhang et al., 2023a*; *Li, 2023*). However, excessive concentrations can severely damage cell membranes, compromise organelle integrity, and threaten plant health. Plants have evolved mechanisms to effectively handle ROS. These mechanisms include enzymatic and nonenzymatic systems that act as antioxidants, neutralize excess ROS, and mitigate their harmful effects, which can help plants maintain this delicate balance (*Maruta et al., 2016*; *Mhamdi & Van Breusegem, 2018*). Removal of excess reactive oxygen species involves various enzymatic scavengers, ascorbate peroxidase (APX), peroxidase (POD), superoxide dismutase (SOD), catalase (CAT), and other antioxidant enzymes (*Faize et al., 2011*). APX exhibits exceptional efficacy in scavenging hydrogen peroxide and superoxide radicals and displays the highest affinity among hydrogen peroxide metabolic enzymes (*Huang et al., 2018*).

APX is a key enzyme involved in removing excess reactive oxygen species. Based on the subcellular localization of its proteins, APX can be categorized into cytoplasmic APX, chloroplast APX, mitochondrial APX, and peroxisomal APX (*Li, 2023*). It encoded by members of a polygenic family. For example, The *APX* gene family in *Arabidopsis* consists of eight members distributed as follows: three in the cytoplasm (*AtAPX1*, *AtAPX2*, and *AtAPX6*), three peroxisomal APX (*AtAPX3*, *AtAPX4*, and *AtAPX5*), and two located in the chloroplasts (*sAPX* in both mitochondria and chloroplasts, *tAPX* in the thylakoid membrane). With the recent advances in plant genome sequencing technology, genome-wide identification and functional analysis of the *APX* gene have been accomplished. To date, the *APX* gene has been identified in many plants. Examples include pepper (*Capsicum annuum* L.) (*Pang et al., 2023*), tomato (*Solanum lycopersicum* L.) (*Hu et al., 2021*), upland cotton (*Gossypium hirsutum* L.), winter rapeseed (*Brassica rapa* L.) (*Ma et al., 2022*), *Arabidopsis thaliana* (*Kameoka et al., 2021*), and sorghum (*Sorghum bicolor* L.) (*Akbudak et al., 2018*). Additionally, various *APX* members have been recognized for their roles in abiotic stress responses. For instance, a significant role of the *APX* gene under salt and drought stress has been demonstrated in both *Arabidopsis* and sorghum. Under these conditions, the expression of *AtAPX1* was significantly upregulated (*Zhang et al., 2023b*). In sorghum, nine *SbAPX* genes were identified, and quantitative real-time PCR (qRT-PCR) results indicated a significant upregulation of *SbAPX* under drought stress (*Akbudak et al., 2018*). The crucial role of the *APX* gene under oxidative and low-temperature stresses has been demonstrated in various plant groups. For example, research on the antioxidant activity and cold resistance of transgenic cassava plants overexpressing *MeAPX2* has revealed that an increase in *APX* expression can effectively eliminate reactive oxygen species, activate antioxidant defense mechanisms, and ultimately bolster resistance to cold stress (*Xu et al., 2014*). Transgenic plants, characterized by the overexpression of *SbpAPX*, demonstrated increased tolerance to both drought and salt stress compared to wild-type plants (*Singh, Mishra & Jha, 2014*). A separate study demonstrated the overexpression of *BcAPX* genes in transgenic Arabidopsis, resulting in higher *APX* expression and activity compared to wild-type plants under heat stress (*Chiang et al., 2015*).

Hemp (*Cannabis sativa* L.) is an important herb that is widely distributed and exhibits strong growth adaptability. In China, hemp has served as a source of textile fibers and has been used in folk medicine since ancient times (*Xie et al., 2023*). Additionally, owing to its richness in lipids, cannabinoids, and flavonoids, it serves as a natural source of phytochemicals (*Cerino et al., 2021*; *Xu et al., 2022*). However, recently, various environmental stressors have threatened the growth, yield, and quality of hemp plants. For instance, the most common abiotic stressors such as drought, low temperature, salinity, and oxidative conditions can inhibit the fluidity of plant cell membranes and induce osmotic stress and reactive oxygen species production, ultimately inhibiting hemp growth. Therefore, exploring the anti-stress mechanisms of hemp is crucial for the further development and utilization of hemp resources, as well as for improving its quality and yield.

This study used datasets containing the whole-genome sequences of hemp and information on the *APX* gene family in *Arabidopsis*, rice, and corn. Bioinformatics analysis techniques were used to identify and analyze the members of the *APX* gene family in hemp. qRT-PCR was used to analyze the expression patterns of *CsAPX* genes under abiotic stress. This study lays the groundwork for further functional analyses of *CsAPX* genes and enhances our understanding of the molecular mechanisms regulating *CsAPX* genes in response to abiotic stress in hemp.

## MATERIALS & METHODS

### Identification of the APX family in hemp

Eight known *AtAPXs* protein sequences were retrieved from the *Arabidopsis* Information Resource database (TAIR, http://www.arabidopsis.org/). These sequences were used as queries in a BLASTP search (E $\leq$ 1e −5) using TBtools to identify *APX* family members within the hemp genome available in the NCBI database (https://www.ncbi.nlm.nih.gov/) (*Chen et al., 2020*). Redundant hemp protein sequences were manually eliminated, and candidate protein sequences containing complete APX domains (PF000141) were subsequently validated as definitive APX protein sequences using Pfam (http://pfam.xfam.org/) (*Mistry et al., 2021*). Gene chromosomal locations were determined using TBtools (https://github.com/CJ-Chen/TBtools-II) (*Chen et al., 2020*). Eight *CsAPXs* were designated based on their chromosomal positions. The molecular weights, isoelectric points, and amino acid counts of the identified sequences were predicted using the ExPASY ProtParam tool (http://Web.ExPASY.Org/protparam/) (*Duvaud et al., 2021*). The subcellular localization of *CsAPX* protein sequences was determined using Cell-PLoc2.0 (http://www.csbio.sjtu.edu.cn/bioinf/Cell-PLoc-2/) (*Chou & Shen, 2008*).

### Multiple sequence alignment and phylogenetic analysis of *CsAPXs*

The sequences used for the interspecific covariance homology relationships between hemp, rice, corn, and Arabidopsis and the corresponding APX sequence information are available in the rice (http://rice.plantbiology.msu.edu), corn (https://www.maizegdb.org/), and TAIR databases (https://www.arabidopsis.org/). Multiple sequence alignments of *APX* genes were conducted using Clustal X2. We constructed a neighbor-joining (NJ) phylogenetic

tree of the relicates by analyzing the protein sequences of CsAPXs, AtAPXs, ZmAPXs, and OsAPXs. Statistical analysis (bootstrap) was performed using MEGA 6.0 (*Zhan et al., 2023*).

## Gene structure, conserved domains and conserved motifs analysis of *CsAPXs*

Genome annotation files (GFF) and coding sequences (CDS) were downloaded from the NCBI database. Conserved domain analysis was conducted using the Conserved Domain Database (CDD) of NCBI, and conserved motif analysis was conducted using MEME online software (https://meme-suite.org/meme/) (*Nystrom & McKay, 2021*). TBtools were used to combine the conserved domains, conserved motifs, gene structure, and gene family evolutionary tree to elucidate the structural and evolutionary relationships within the gene family (*Chen et al., 2020*).

## Genome collinearity analysis and chromosomal localization of *CsAPXs*

A chromosome distribution map of the *CsAPX* genes was drawn using the TBtools. Genome collinearity analysis was also conducted using Advanced Circos from the TBtools software with default settings (*Chen et al., 2020*). One Step MCScanX was used to estimate the non-synonymous substitution rate (Ka), synonymous substitution rate (Ks), and the ratio (Ka/Ks) of each paralog pair (*Wang et al., 2010*).

## Analysis of Cis-Acting elements of *CsAPXs*

The 2000-bp sequence upstream of the nucleotide "A" from the translation initiation site (ATG) was extracted to serve as the promoter sequence for *CsAPXs*. PlantCARE (http://bioinformatics.psb.ugent.be/webtools/plantcare/html/) was then used to predict potential cis-acting elements within the promoter sequence (*Passricha et al., 2017*). Subsequently, visualization, classification, and analysis were performed using TBtools (*Chen et al., 2020*).

## Plant material and stressful conditions

In the present study, the hemp cultivar "Qingma 3" was used. For expression analysis of *CsAPXs* under cold stress, seeds were planted in a mixture of peat moss and perlite at a ratio of 3:1 in a nutrient bowl. For expression analysis under drought, salt, and oxidative stresses, hemp was cultivated in a hydroponic box. These plants were cultivated in a growth chamber under controlled conditions with a day/night temperature regimen of 24 °C/16 °C and a photoperiod of 16 h of light and 8 h of darkness. For cold stress treatment, four-week-old seedlings cultured in growth chambers were placed in the 4 °C climate chamber. For salt, oxidative, and drought stress treatments, the seedlings were treated with Hoagland's nutrient solution enriched with 250 mmol/L NaCl, 10 mmol/L $H_2O_2$, and 20% PEG6000, respectively. Leaf samples were collected at 0, 3, 6, 12, 24, and 48 h after treatment, with approximately 100 mg of whole blades of true leaves collected at each time point. All treatments were performed in triplicate. Subsequently, all samples were promptly frozen in liquid nitrogen and stored at −80 °C for RNA extraction.
### RNA extraction and qRT-PCR analysis

We systematically selected all *CsAPXs* for qRT-PCR. Total RNA was extracted from hemp plant samples using the TRIzol method, and first-strand cDNA synthesis was performed using the SureScript™ First-Strand cDNA Synthesis Kit (Guangzhou, China). Specific primers were meticulously designed using the Primer Premier 5.0 software and then synthesized by a reputable source designated as Sangon Biotech Co, Ltd. Primer specificity was verified using NCBI Primer BLAST. Comprehensive primer sequence information is available for reference (Table S1). The BlazeTaq™ SYBR Green qPCR Mix 2.0 reagent from GeneCopoeia Company was used. The system was set up with a 10 μL configuration, following the provided instructions, and qRT-PCR analysis was subsequently conducted using the America BIO-RAD CFX90 quantitative PCR instrument. Detailed procedures: Stage 1, 95 °C 30 s for 1 cycle; Stage 2, 95 °C 10 s, 60 °C 30 s, 65 °C 10 s for 40 cycles. A minimum of three biological replicates were established for each sample, and the relative expression levels of individual genes were determined using the $2^{-\Delta\Delta CT}$ method. The hemp Actin gene was used as the reference gene, and this reference gene has been shown in the hemp genome to be the most stable (*Yan et al., 2023*). Statistical analysis involved the use of GraphPad Prism 8 to assess variance, and significance was evaluated through one-way ANOVA, supplemented by nonparametric or mixed models where appropriate.

## RESULTS

### Identification of *APX* genes in hemp and analysis of their physicochemical properties

Using TBtools for analysis and HMMER and SMART online tools for screening, we analyzed eight *CsAPXs* and named them based on their positions on chromosome *CsAPX1-8*. Comprehensive details regarding *CsAPXs* and the physicochemical characteristics of their encoded proteins are presented in Table 1.

In the *CsAPX* gene family, the coding sequence of *CsAPX6* was the longest, spanning 1,926 bp, whereas *CsAPX7* had the shortest coding sequence, consisting of 750 bp; the average length was 1,011 bp. The distribution of these eight *CsAPX* family gene members is uneven across the five hemp chromosomes. The physicochemical properties of the *CsAPXs* proteins were determined using ExPASy online tools. The lowest molecular weight observed was 27.17 kDa for *CsAPX7*, whereas the highest was 71.59 kDa for *CsAPX6*. The aliphatic indices ranged from 35.65 (*CsAPX2*) to 52.18 (*CsAPX8*). The instability index exhibited a wide range of values, with *CsAPX2* having the lowest value (35.65) and *CsAPX8* having the highest value (52.18). Similarly, the theoretical isoelectric point varied across *CsAPXs* proteins, with *CsAPX7* having the lowest value (5.54) and *CsAPX4* having the highest value (8.87). Hydrophilicity and hydrophobicity analyses indicated that all *CsAPXs* proteins exhibit hydrophilic properties. Surfactant proteins are directly exposed to the organellar environment, which is determined by the amino acid sequence and depends on the amino acid composition. Bioinformatic predictions of subcellular localization can therefore be made from amino acid composition. The prediction of subcellular localization using online software revealed that out of the eight members of the *CsAPX* gene family,

**Table 1** Protein properties about the eight *CsAPXs*.

| Gene Name | Gene ID | CDS Length (bp) | Chr | Protein Characteristics | | | | | Subcellular Location |
|---|---|---|---|---|---|---|---|---|---|
| | | | | Theoretical pI | MW (kDa) | Aliphatic Index | Instability Index | GRAVY | |
| *CsAPX1* | XP_030485953.1 | 845 | 1 | 8.27 | 31.11 | 86.80 | 40.66 | −0.391 | Peroxisome |
| *CsAPX2* | XP_030485951.1 | 869 | 1 | 5.88 | 31.58 | 87.75 | 35.65 | −0.290 | Peroxisome |
| *CsAPX3* | XP_030487766.1 | 866 | 1 | 7.74 | 31.72 | 78.96 | 39.94 | −0.392 | Peroxisome |
| *CsAPX4* | XP_030491786.1 | 1302 | 1 | 8.87 | 47.33 | 75.36 | 44.29 | −0.407 | Cytoplasm |
| *CsAPX5* | XP_030493443.1 | 758 | 3 | 5.87 | 27.46 | 80.76 | 36.06 | −0.433 | Chloroplast/mitochondrion |
| *CsAPX6* | XP_030496013.1 | 1926 | 4 | 8.70 | 71.59 | 78.49 | 40.36 | −0.341 | Cytoplasm |
| *CsAPX7* | XP_030504263.1 | 750 | 6 | 5.54 | 27.17 | 83.49 | 37.10 | −0.283 | Peroxisome |
| *CsAPX8* | XP_030490534.1 | 771 | 9 | 8.77 | 15.10 | 84.86 | 52.18 | −0.251 | Peroxisome |

five were predicted to be localized to peroxisomes, two were predicted to be localized in the cytoplasm, and one was predicted to be localized in the chloroplasts/mitochondria.

## Phylogenetic relationship analysis of *CsAPXs*

To investigate the phylogenetic relationships between *APX* genes and their homologous counterparts, a phylogenetic tree was constructed using the protein sequences of APX in *Arabidopsis*, hemp (*Cannabis sativa* L.), rice (*Oryza sativa* L.) and corn (*Zea mays* L.) (Fig. 1). Based on the subcellular localization, the phylogenetic tree was divided into three subfamilies that have been reported previously: perAPXs (peroxisomal APXs subfamily), cytAPXs (cytoplasmatic APXs subfamily), and orgAPXs (organellar APXs subfamily) (*Jardim-Messeder et al., 2023*). However, bootstrap values for the AtAPX4, AtAPX6, and CsAPX6 clades clearly indicated that these three APXs are not part of any subfamilies; they belong to the subfamilies APX-R (AtAPX6 and CsAPX6) and APX-L (AtAPX4) (*Lazzarotto et al., 2011*; *Lazzarotto et al., 2021*; *Lazzarotto, Turchetto-Zolet & Margis-Pinheiro, 2015*). To focus on APX genes, APX-R and APX-L were completely separated from APX genes. Therefore, AtAPX4, CsAPX6, and AtAPX6 were selected for deletion based on the phylogenetic tree. The orgAPXs subfamily encompassed ten APXs, including three CsAPXs (CsAPX5, CsAPX7, and CsAPX8). Similarly, the perAPXs subfamily comprised nine APXs, including three CsAPXs (CsAPX1, CsAPX2, and CsAPX3), whereas the cytAPXs subfamily consisted of ten genes, including only CsAPX4.

## Gene structure, conserved domains and conserved motifs analysis of *CsAPXs*

We analyzed the conserved motifs, conserved domains, and gene structures of *CsAPXs*, as illustrated in Fig. 2. In this motif analysis, nine conserved motifs were identified and labeled motifs 1–9. The motifs in the same group exhibited a high degree of similarity. All *CsAPXs* contained motif 1, motif 2, motif 3, and motif 4. This observation suggests that these motifs serve as conserved elements that represent functional motifs within the *CsAPX* gene family. Other *CsAPXs* except *CsAPX6* contained conserved motifs 5 and 8. Motif 7 was present only in *CsAPX2* and *CsAPX3* and the motifs were consistently arranged in a uniform order. In the analysis of conserved domains, we identified three highly conserved

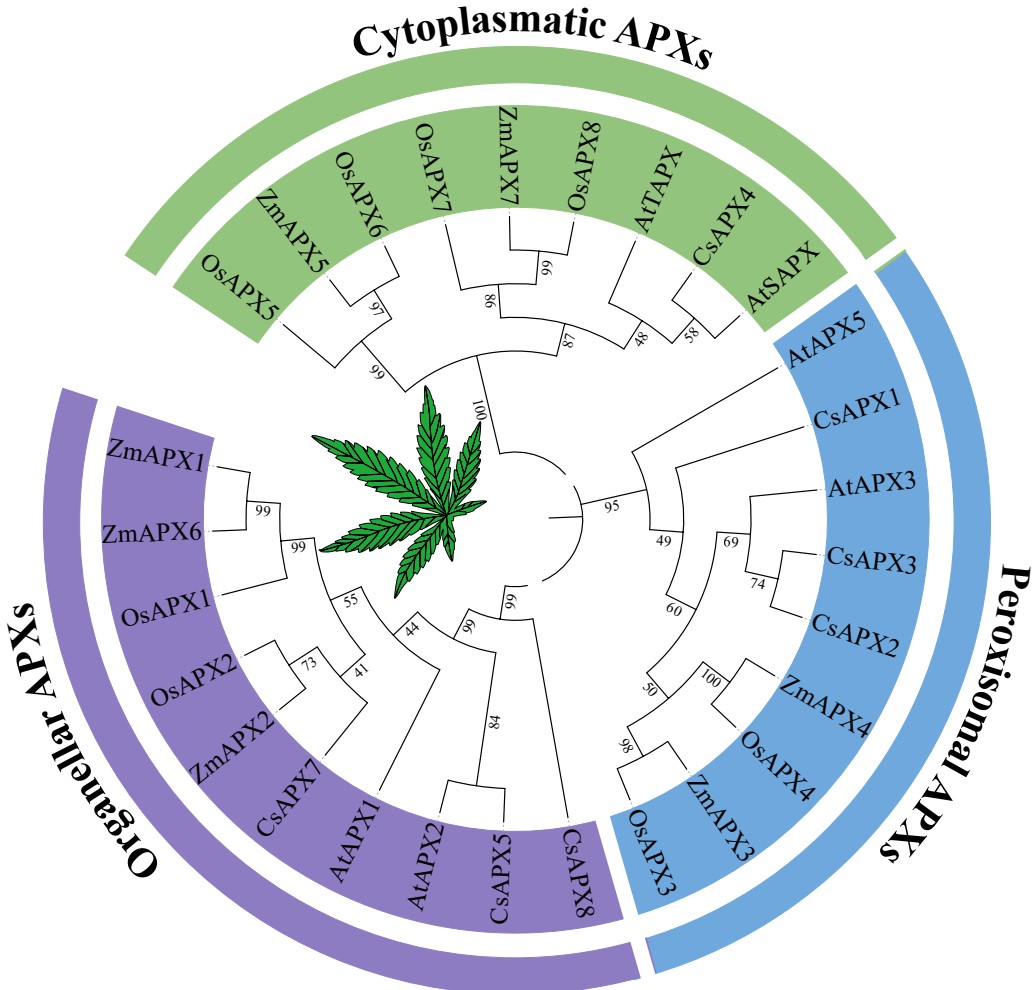

**Figure 1 Phylogenetic tree analysis of the APX in *Arabidopsis thaliana*, hemp (*Cannabis sativa* L.), rice (*Oryza sativa* L.) and corn (*Zea mays* L.).** The phylogenetic relationships between APX genes and their homologous counterparts. There are six AtAPXs, seven CsAPXs, eight OsAPXs, and seven ZmAPXs. The colors indicate three clusters that represent Organeller APXs (purple), Peroxisomal APXs (blue), and Cytoplasmatic APXs (green).

domains in the members of the *CsAPX* gene family. Domains in the same group were highly similar. Motifs 1, 2, and 3 harbor a PLN-conserved domain located between amino acids 9 and 289, classifying them as members of the ascorbate peroxidase (APX) superfamily (PF00141). Plant peroxidase and ascorbate peroxidase conserved domains are also closely related to peroxidases in plants. Gene structure analysis revealed that the number of exons ranged from nine to 12. Introns varied in length, with eight to 11 introns per gene. Six *CsAPXs* harbored nine exons, comprising 75% of the family members, whereas five *CsAPXs* featured eight introns, accounting for 62.5% of the total *CsAPXs*. *CsAPX4* in the cytAPXs subfamily contains nine exons and eight introns with different exon-intron structures.
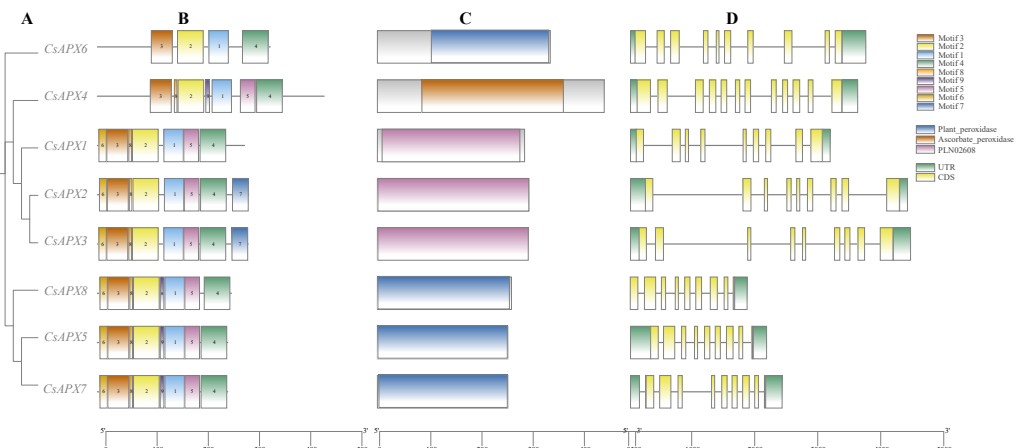

**Figure 2** **Phylogenetic relationships, gene structure, conserved domains, and conserved motifs analysis of *CsAPXs*.** (A) Phylogenetic tree of all CsAPX proteins constructed using maximum-likelihood method; (B) Motif distribution of CsAPXs; Motifs 1–9 are shown as rectangular boxes of different colors; (C) Conserved domain distribution of CsAPX proteins; the brown box indicates the Plant-peroxidase domain in the corresponding amino acid sequence, the yellow box indicates the Ascorbate-peroxidase domain in the corresponding amino acid sequence, the purple box indicates the PLN02608 domain in the corresponding amino acid sequence; (D) Gene structures of CsAPXs arranged according to phylogenetic relationship; green boxes represent 5′ UTR and 3′ UTR, yellow boxes represent exons, and gray lines represent introns.

## Genome collinearity analysis and chromosomal localization of *CsAPXs*

The correlation between the locations of *CsAPX* on the chromosomes is shown in Fig. 3. *CsAPXs* were unevenly distributed on h chromosomes of hemp, with eight genes distributed across five hemp chromosomes. Among the eight *CsAPXs*, *CsAPX4* and *CsAPX5* were located in regions with low gene density. Remarkably, chromosome 1 contained the largest number of *CsAPXs*, encompassing four genes, whereas chromosomes 3, 4, 6, and 9 contained a single *CsAPX* gene. Small gene clusters were found on chromosome scaffold 1 based on the definition of gene clusters.

Examination of collinearity within the species revealed duplication relationships among genes, which are illustrated within the same species in Fig. 4. *CsAPXs* have only one collinear relationship within the species, namely *CsAPX1* with *CsAPX6*.

Collinearity analysis was conducted to ascertain the collinear relationships among the eight *CsAPXs* in comparison with other species. Arabidopsis and corn were included in this analysis along with *APX* gene family in hemp. The results of this analysis are shown in Fig. 5. Hemp exhibited four collinear relationships with *Arabidopsis* and only one with corn. Among the *CsAPXs* in hemp, *CsAPX7* exhibited a close relationship with the other two species. Comparative analysis of the *APX* gene family across different species can provide a valuable reference for investigating genetic relationships and functions within species.
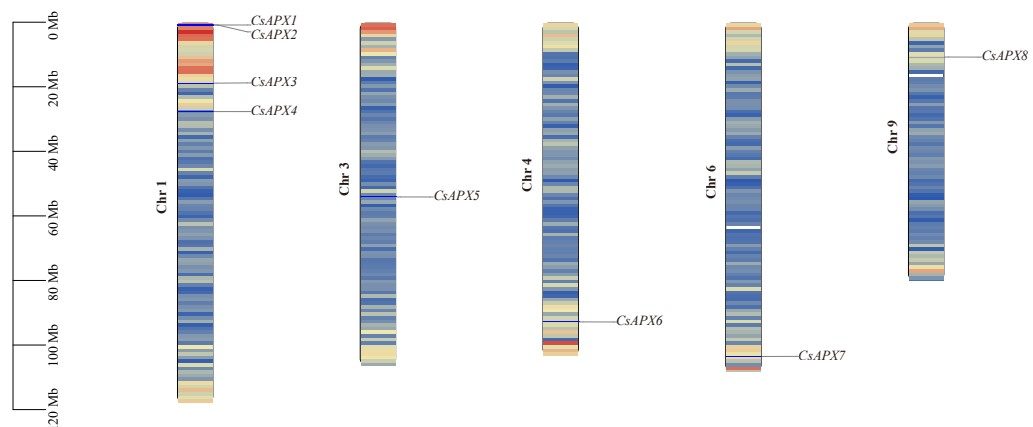

**Figure 3** **Distribution of *CsAPXs* on the *Cannabis sativa* chromosomes.** The scale on the left side estimates chromosome length. Members of the *CsAPX* gene family are sequentially numbered from 1 to 8, indicated in black. The black values on the chromosome's left side represent the chromosome numbers. A color gradient from blue to red visually depicts gene density on the chromosomes.

## Analysis of Cis-acting elements of *CsAPXs*

Cis-acting elements play significant roles in regulating gene transcription. To investigate the presence of cis-acting elements in the *CsAPX* gene family, a 2,000-bp sequence upstream of each gene was analyzed. The results depicted in Fig. 6 revealed 29 cis-acting elements. These elements were further classified into three distinct groups: plant growth and development elements (71), phytohormone-responsive elements (82), and abiotic and biotic stress elements (109), resulting in 262 predicted cis-acting elements. It is worth noting that all *CsAPXs*, except *CsAPX3*, contain Box 4, a cis-acting element that can transmit signals in a circuit and amplify or attenuate them.

Among the *CsAPXs*, only *CsAPX1* contained an MRE element (a cis-acting element that responds to anoxia). Similarly, all members of the *CsAPX* gene family except *CsAPX1* contain an MYB element. Additionally, the CARE element was found in *CsAPX4* only. Analysis of the promoter region of the *CsAPX* gene family members revealed that *CsAPX5* had the highest number (17) of cis-acting elements associated with abiotic and biotic stresses, whereas *CsAPX1* had the lowest count. *CsAPX5* also had the highest number (17) of phytohormone-responsive elements, whereas *CsAPX2* had the lowest count. Furthermore, *CsAPX7* had the highest number (16) of plant growth and development elements, whereas *CsAPX3* had the lowest count. The cis-acting elements identified in the promoter region enhance our understanding of the expression patterns of the *CsAPX* gene family.

## Expression analysis of the *CsAPXs* under abiotic stresses

We investigated the potential involvement of *CsAPXs* in the response to various stresses, such as cold, drought, salt, and oxidative stress by analyzing their relative expression levels in hemp leaves subjected to these stresses at various time points (0, 3, 6, 12, 24, and 48 h). The results are presented in Fig. 7.

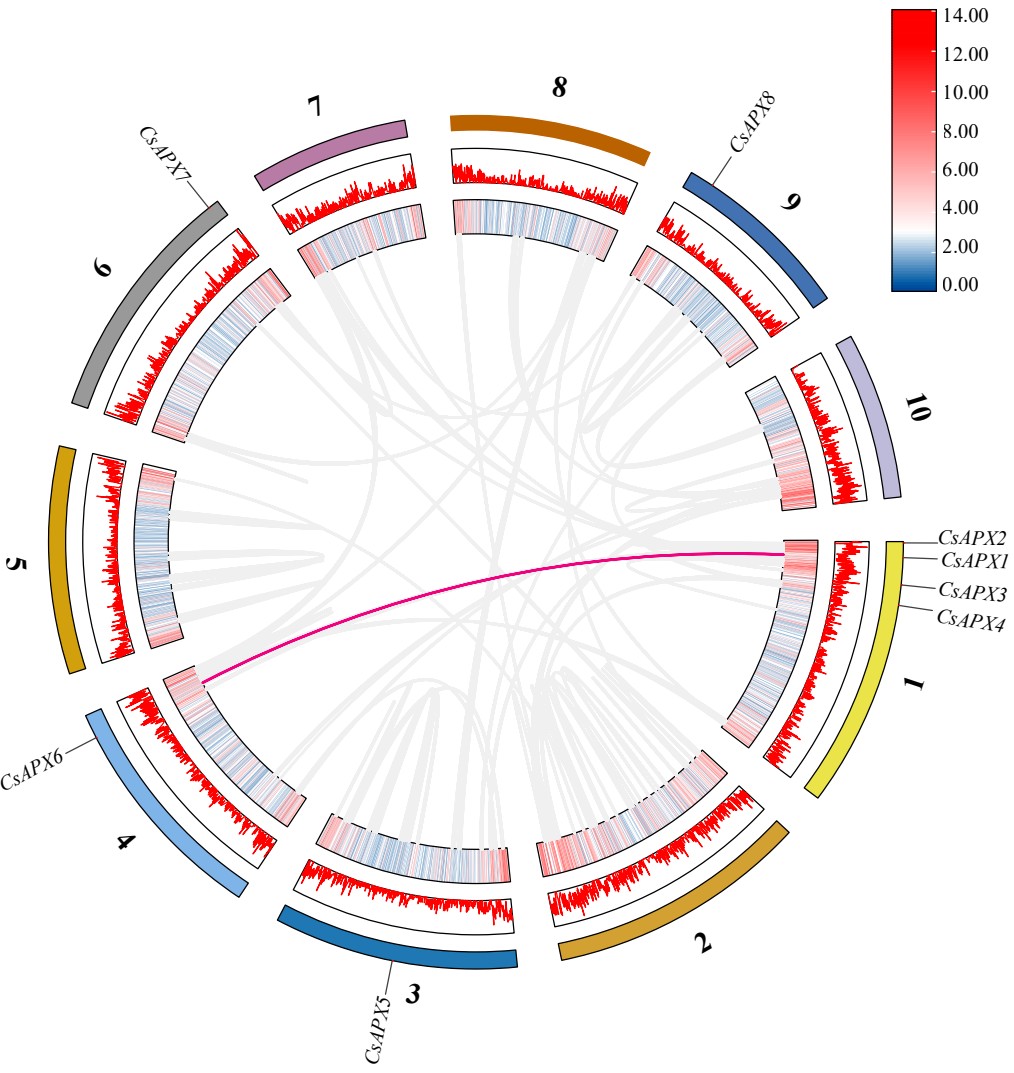

**Figure 4   Collinearity analysis of *CsAPX* gene family members.** From the outside in, different colors represent chromosomes 1-10, the red line indicates the genome's GC ratio, and the innermost circle's heatmap shows gene density. In the center, gray lines represent synteny blocks in the *Cannabis sativa* genome, and pink lines between chromosomes mark segmental duplication gene pairs.

During cold treatment, all members of the *CsAPX* gene family except *CsAPX5* exhibited significant upregulation initially as downregulation followed by subsequent upregulation. Notably, variations in the extent of the upregulation and response times were observed (Fig. 7A). Approximately 50% of *CsAPXs* exhibited the most pronounced response at 48 h. Among these, *CsAPX1* displayed the most robust response to cold stress, with an upregulation exceeding 20-fold. Similarly, *CsAPX2*, *CsAPX3*, and *CsAPX7* exhibited strong responses, with an upregulation exceeding 10-fold.

During drought treatment (Fig. 7B), *CsAPX4*, *CsAPX5*, and *CsAPX8* were significantly downregulated within the first 6 h, in contrast to the induction observed for other *CsAPXs*

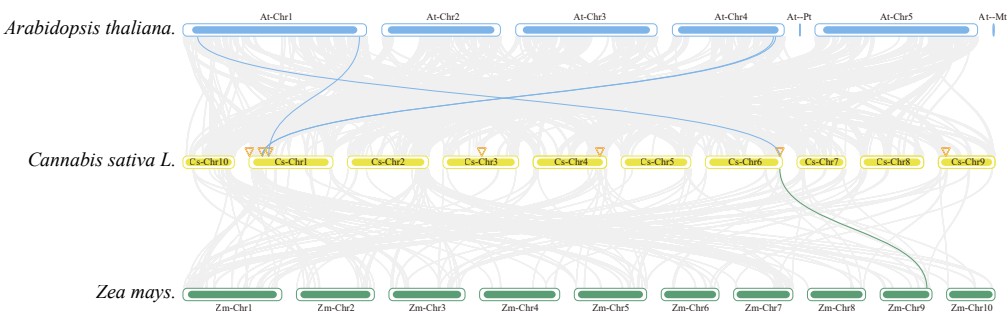

**Figure 5** Interspecific collinearity relationships of the *CsAPX* genes among *Arabidopsis*, hemp (*Cannabis sativa* L.), and corn (*Zea mays* L.). Homologous *APX* gene pairs exist between Arabidopsis and hemp (*AtAPX1-CsAPX7, AtsAPX-CsAPX4, AtAPX3-CsAPX3*, and *AtAPX5-CsAPX3*), and between hemp and corn (*CsAPX7-ZmAPX2*). Chromosomes from Arabidopsis (blue), hemp (yellow), and corn (green) are color-coded for distinction, with chromosome numbers indicated either above, below, or inside each chromosome. Gray lines represent hemp gene blocks orthologous to other genomes, while blue and green lines delineate syntenic APX gene pairs. The location of *CsAPX* genes is marked by a yellow triangle.

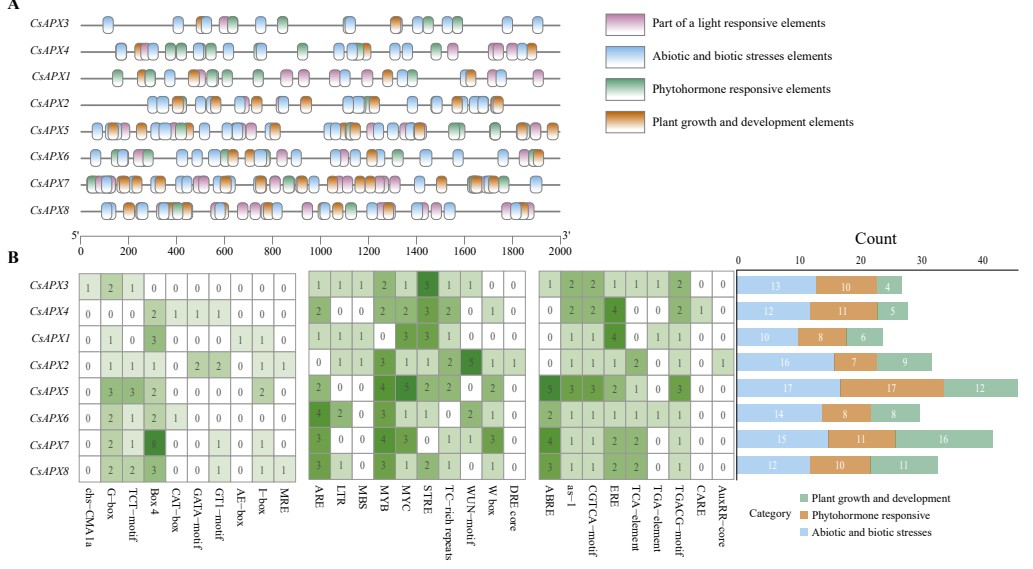

**Figure 6** Analysis of cis-acting elements in the promoter region of *CsAPX* genes. (A) The distribution of cis-acting elements in the promoter region of *CsAPX* genes (−2000 bp) is represented by different colors. The ruler at the bottom indicates the sequence's direction and length; (B) The classification and statistical analysis of cis-acting elements are presented. Twenty-nine types of cis-acting elements, excluding light-responsive elements, are categorized into three groups: abiotic and biotic stress elements (blue), phytohormone responsive elements (brown), and plant growth and development elements (green). In the left grid, each element's count is represented numerically, with a color gradient from white to green indicating an increase in element count. The bar chart on the right counts the total number of different categories of cis-acting elements in each *CsAPX* gene.

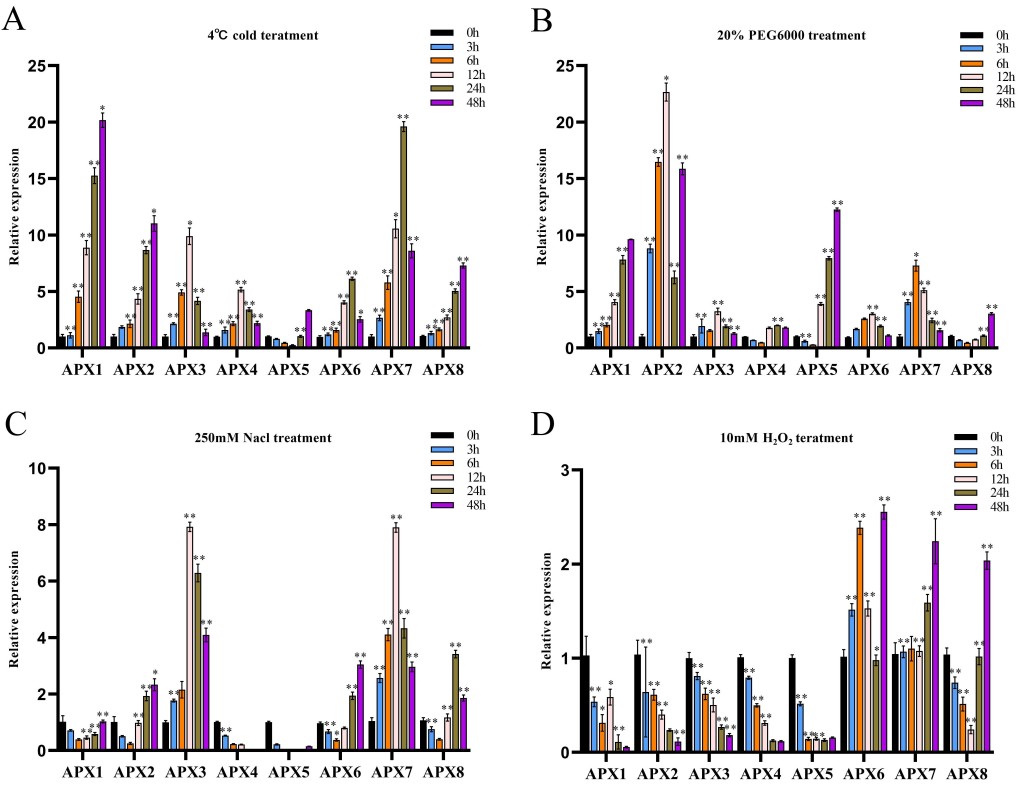

**Figure 7** **The relative expression levels of *CsAPX* genes were evaluated under various stress conditions: cold stress (A), drought stress (B), salt stress (C), and oxidative stress (D), at time points of 0, 3, 6, 12, 24, and 48 hours.** Values are presented as the mean ± standard deviation (SD) of three independent biological replicates, relative to the 0-hour control. Asterisks indicate statistical significance compared to the control (0 h), as determined by a two-tailed Student's $t$-test. Asterisks indication of statistical significance: ** $p < 0.01$, and * $p < 0.05$.

under the same stress. *CsAPX3* and *CsAPX6* exhibited similar response patterns with peaks at 12 h. Conversely, *CsAPX1*, *CsAPX5*, and *CsAPX8* shared a common response mode, with the maximum expression observed at 48 h. Importantly, *CsAPX7* reached its highest responsiveness at 6 h, whereas *CsAPX2* displayed the most substantial upregulation of approximately 22-fold at 12 h.

In response to salt treatment (Fig. 7C), the relative upregulation of *CsAPXs* was less pronounced than their responses to drought and cold treatments. Apart from *CsAPX4* and *CsAPX5*, which were significantly downregulated, the rest of the members of the *CsAPX* gene family were upregulated. Notably, there were variations in the degree of upregulation and response times. *CsAPX3* and *CsAPX7* exhibited similar response patterns, reaching their peak at 12 h, with upregulation exceeding eight-fold.

In response to oxidative stress (Fig. 7D), the upregulation of *CsAPXs* was relatively less pronounced than their responses to salt stress. Among the *CsAPX* gene family members, only *CsAPX6*, *CsAPX7*, and *CsAPX8* were significantly upregulated in response to oxidative stress, whereas the other members were repressed. Notably, we observed variations in the

degree of upregulation and response times. *CsAPX6*, *CsAPX7*, and *CsAPX8* reached their maximum expression levels at 48 h, with *CsAPX6* displaying the most robust response to oxidative stress, exceeding a 2.5-fold increase.

## DISCUSSION

### *CsAPX* gene family identification and phylogenetic relationships

DNA sequencing data from the hemp genome provide valuable insights into the hemp gene family and enhance our understanding of gene functions and regulatory mechanisms. This study identified eight members of the *APX* gene family in hemp, which is equal to the number in *Arabidopsis thaliana* but greater than that in *Oryza sativa* (*Huang et al., 2018*; *Kameoka et al., 2021*). This study identified eight *CsAPXs* within the hemp genome, which were classified into four distinct subfamilies: perAPXs, orgAPXs, cytAPXs, and APX-R. However, the bootstrap values for the CsAPX6 clades clearly indicated their affiliation with an independent APX-R subfamily. The APX-R subfamilies will not be discussed further here, as they have been completely separated from the APX gene family (*Lazzarotto et al., 2011*; *Lazzarotto et al., 2021*). Subcellular localization predictions suggest that the expression sites of *APX* genes in hemp are similar to those in *Arabidopsis thaliana* and *Oryza sativa*, with most *APX* gene family members localized in peroxisomes. Phylogenetic analyses revealed that all members (*CsAPX5*, *CsAPX7*, and *CsAPX8*) of the orgAPXs subfamily are located in organelles (chloroplasts/mitochondria). However, subcellular localization predictions indicated that *CsAPX7* and *CsAPX8* were located within the peroxisomes, which was inconsistent.

Bioinformatic predictions of subcellular localization are based on amino acid composition, which in turn depends on the amino acid sequence. Thus, phylogenetic analyses based on amino acid sequences are more reliable than bioinformatic predictions for predicting subcellular localization. *CsAPX7* and *CsAPX8* are expected to be located in the same organelles as *CsAPX5*. However, for the orgAPXs and perAPXs subfamilies, phylogenetic analyses and subcellular localization predictions were in perfect agreement. *CsAPX4*, a member of the cytAPXs subfamily, is found in the cytoplasm, whereas all members of the perAPXs subfamily are located in the peroxisomes. This suggests a potentially shared subcellular localization among close relatives, as shown in Fig. 1. *CsAPX5* from the orgAPXs subfamily was consistently found within both the chloroplasts and mitochondria, corroborating earlier findings on *OsAPX5* in rice and *sAPX* in *Arabidopsis* (*Ribeiro et al., 2017*; *Kameoka et al., 2021*).

*CsAPXs* located in peroxisomes exhibit seven distinct conserved motifs and possess nine exonic regions, demonstrating the functional diversity of the *APX* family in response to various stress conditions. Furthermore, different *APX* subfamilies exhibit low homology and distinct functional domains. This observation suggests significant divergence in the origin and functional evolution within the *CsAPX* gene families. The 26 *GhAPXs* discovered in cotton were categorized into five subfamilies, with notably low homology observed among these distinct *APX* subfamilies (*Tao et al., 2018*). In another study, construction of a phylogenetic tree encompassing *AtAPXs* and *SbAPXs* sequences revealed

that diverse clusters of APX proteins contain distinct subcellular localizations, and it was inferred that *SbAPXs* exhibit functions analogous to *Arabidopsis* counterparts within the corresponding cluster (*Akbudak et al., 2018*; *Kameoka et al., 2021*). Rice and corn are monocotyledonous plants, whereas *Arabidopsis* and hemp are dicotyledonous plants. Consequently, a phylogenetic tree encompassing hemp, *Arabidopsis*, rice, and corn was constructed, revealing a close evolutionary relationship between *CsAPXs* and *APX* genes in other species.

## Genome collinearity and chromosomal localization analysis of *CsAPXs*

Gene replication events occur frequently in plant genomes. This process has contributed to the efficient evolution of plants (*Li et al., 2023*). Members of a gene family share a common ancestor and evolve through mutation, domestication, and selection, resulting in the formation of a cluster of genes that encode similar proteins with comparable sequences and structures. These events can stem from gene replication or copy generation *via* whole-genome duplication (WGD) (*Murat, Peer & Salse, 2012*). Intraspecific collinearity analysis, as depicted in Fig. 4, revealed collinearity solely between *CsAPX1* and *CsAPX6*, suggesting that *CsAPX* gene family members either had fewer copies or experienced more frequent copy loss during evolution. To further explore the homologous relationships among *APX* gene family members in plants, we conducted an interspecific collinearity analysis involving *Arabidopsis*, hemp, and corn. Notably, collinearity analysis revealed that hemp exhibited four collinearities with *Arabidopsis* and only one with corn, as shown in Fig. 5. This finding is consistent with the genetic distance (*Qiao et al., 2019*). *CsAPX1*, *CsAPX5*, *CsAPX6*, and *CsAPX8* lacked collinear relationships with the other two species, suggesting their potential uniqueness to hemp. In contrast, the remaining members of the *CsAPX* gene family exhibited collinear relationships with corn and *Arabidopsis*. Furthermore, we observed that the eight *CsAPXs* exhibited an uneven distribution across the five hemp chromosomes (Fig. 3). Notably, all *CsAPXs*, except *CsAPX5*, were located at the distal end of the chromosome.

## Gene structure and Cis-Acting elements analysis of CsAPXs

In the *CsAPXs* conserved motif analysis, all eight *CsAPXs* displayed motifs 1, 2, 3, and 4 of the nine conserved motifs. All genes except *CsAPX6* contained at least seven conserved motifs in a consistent order. *CsAPX4*, a gene with at least seven conserved motifs, has not been linked to specific functions. The orgAPXs and perAPXs subfamilies are believed to have evolved from this gene, which may explain the incomplete nature of their conserved motifs (*Sun et al., 2023*). Identifying conserved motifs in the *CsAPX* gene family is crucial for gene classification and functional prediction. The typical structure of a gene in the *CsAPX* family comprises nine exons. Approximately 75% of the family members share this structure, although some may have up to 11 introns. *CsAPX4* has 12 exons, which is more than that of other members of the *CsAPX* gene family. Moreover, the conserved motif analysis was also in good agreement with its gene structure; therefore, *CsAPX4* gene is likely to have evolved from two genes spliced together.

Promoter cis-acting elements play a crucial role in regulating various plant growth and development processes, including exogenous hormone induction and responses to
abiotic stress (*Yang et al., 2019*). During protein transcription, the core promoter of a gene typically features a "TATA-box". Both the cis-acting element and core promoter can serve as transcription binding sites and regulate specific protein-binding sites (*Koul et al., 2019*; *Rehman et al., 2022*).

### The response of *CsAPXs* to four various abiotic stresses

To investigate the relationship between the *APX* gene family and stress resistance mechanisms in Chinese hemp, we used qRT-PCR to examine the expression patterns of *CsAPXs* under cold, drought, salt, and oxidative stress conditions. *CsAPX* expression patterns varied under four abiotic stress conditions. Cold treatment significantly upregulated *CsAPX1* and *CsAPX7* expression, whereas drought treatment significantly upregulated *CsAPX2* and *CsAPX5* expression. These findings imply that *CsAPX* gene family members could play a vital role in the response of hemp to abiotic stresses, corroborating previous research results (*Liu et al., 2019*). However, some *CsAPXs* are down-regulated in response to abiotic stress. Specifically, under oxidative stress, all members of the *CsAPX* gene family, except for *CsAPX6*, *CsAPX7*, and *CsAPX8,* showed consistently lower expression levels at 48 h. This implies that the expression of these genes is suppressed following stress, and other genes or alternative mechanisms might eliminate reactive oxygen species (ROS) (*Wang, Hecker & Hauser, 2014*).

Prior research has shown that APX-transgenic plants, including *Arabidopsis*, potatoes, sweet potatoes, and cotton, usually have a higher APX content and an improved ability to detoxify ROS under various abiotic stresses (*Guan et al., 2015*; *Yan et al., 2016*; *Shafi et al., 2017*; *Zhou et al., 2021*). In the present study, we noted significant upregulation of *CsAPX6*, *CsAPX7*, and *CsAPX8* under all four abiotic stress conditions, with a particularly marked upregulation under cold and salt stresses. Moreover, the expression patterns of the individual genes showed contrasting trends under different abiotic stress conditions. For example, the expression level of *CsAPX2* significantly increased in response to cold and drought stress, but decreased under salt and oxidative stresses. In contrast, some genes were expressed only in response to specific abiotic stressors. This highlights the complexity of *CsAPX* responses to various abiotic stressors. These findings are consistent with those of previous studies. For example, the expression of *SocAPX* in spinach increased under strong light but its transcription level did not change under drought stress. Furthermore, the transcription levels of *mAPX*, *sAPX*, and *tAPX* remained unchanged after salt, ABA, and drought treatments (*Yoshimura et al., 2000*). Harsh ecological conditions in northeastern China have been subjected to abiotic stresses throughout the entire growth period, including cold, drought, salt, and oxidative stress. This implies that hemp varieties with wide adaptability may exhibit greater resistance than other varieties. Therefore, *APX* genes that respond simultaneously to multiple abiotic stressors warrant further investigation.

## CONCLUSIONS

In the present study, we analyzed the *CsAPX* gene family and identified eight candidate members categorized into three subfamilies. We performed a thorough and systematic bioinformatics analysis of *CsAPX* gene family members. This analysis included an

investigation of the evolutionary relationships, gene structures, conserved motifs, conserved domains, chromosomal locations, intraspecific collinearity, interspecific collinearity, and cis-acting elements. This analysis was performed to enhance our understanding of their potential functions. Additionally, we analyzed the expression patterns of these *CsAPXs* under various abiotic stresses to gain insights into how they regulate the responses of hemp. The results indicated that most genes exhibit distinct expression patterns under various abiotic stresses, suggesting temporal and spatial variation in their functions. In response to cold stress, *CsAPX1* and *CsAPX7* were highly induced. *CsAPX2* and *CsAPX5* were significantly upregulated in response to drought stress. *CsAPX3* and *CsAPX7* were strongly upregulated in response to salt stress. *CsAPX6*, *CsAPX7*, and *CsAPX8* were significantly upregulated in response to oxidative stress. Furthermore, we identified three key genes (*CsAPX6*, *CsAPX7*, and *CsAPX8*) that responded to abiotic stress. These genes are closely related to the response of hemp to abiotic stress, and warrant further investigation. This study offers novel insights into the role of the *APX* gene family in the molecular mechanisms underlying the resistance of hemp to adversity.

### Funding

This work was supported by the Basic Research Operating Expenses of Provincial Undergraduate Universities of Hemp in Heilongjiang Province (145109508, 145209514). The funders had no role in study design, data collection and analysis, decision to publish, or preparation of the manuscript.

### Grant Disclosures

The following grant information was disclosed by the authors:
Basic Research Operating Expenses of Provincial Undergraduate Universities of Hemp in Heilongjiang Province: 145109508, 145209514.

### Competing Interests

The authors declare there are no competing interests.

### Author Contributions

- Zixiao Liang conceived and designed the experiments, performed the experiments, analyzed the data, prepared figures and/or tables, and approved the final draft.
- Hongguo Xu conceived and designed the experiments, authored or reviewed drafts of the article, and approved the final draft.
- Hongying Qi analyzed the data, authored or reviewed drafts of the article, and approved the final draft.
- Yiying Fei performed the experiments, analyzed the data, prepared figures and/or tables, and approved the final draft.
- Jiaying Cui analyzed the data, prepared figures and/or tables, and approved the final draft.

## Data Availability

The raw data is available in the Supplemental File.

## Supplemental Information

Supplemental information for this article can be found online at http://dx.doi.org/10.7717/peerj.17249#supplemental-information.

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
