# Peer review of "Genome-wide identification and analysis of ascorbate peroxidase (APX) gene family in hemp (Cannabis sativa L.) under various abiotic stresses"

_PeerJ, doi:10.7717/peerj.17249_

## Round 0.1 · original submission · Major Revisions

Abiotic stressors stand out as pivotal factors capable of impeding the optimal growth of plants. Every research endeavor aimed at unraveling the intricacies of plant responses to these stressors holds immense value. I am sure that the findings of your study will serve as a crucial resource for future researchers. However, it is essential to address both technical and linguistic aspects in refining your article. I recommend a thorough review of the reviewers' suggestions and a judicious consideration of each recommendation. If you find yourself in disagreement with any particular suggestion, it would be beneficial to provide clear and well-reasoned justifications for your perspective.

**Language Note:** PeerJ staff have identified that the English language needs to be improved. When you prepare your next revision, please either (i) have a colleague who is proficient in English and familiar with the subject matter review your manuscript, or (ii) contact a professional editing service to review your manuscript. PeerJ can provide language editing services - you can contact us at [email protected] for pricing (be sure to provide your manuscript number and title). – PeerJ Staff

Reviewer 1 ·

Basic reporting

In the manuscript (Genome-wide identiûcation and analysis of ascorbate peroxidase (APX) gene family in hemp (Cannabis sativa L.) under various abiotic stresses), the authors mainly studied eight members of the CsAPX gene family in hemp and studied their physicochemical properties. And the expression difference of 8 CsAPX in leaf tissues under cold, drought, salt stress and oxidation stress was determined. However, I think that the manuscript should be revised before publication.

Major:
1. In the abstract, the author introduced the research object and content. However, the introduction of the research purpose is not clear, and the summary of the research conclusions in this paper is not comprehensively enough.
2. In line 284, the author's analysis of the table is not very accurate. Not only was the expression of CsAPX4 significantly down-regulated, but also CsAPX8. At the same time, at the end of this paragraph, the author states that CsAPX2 achieves the most substantial up-regulation in 12 hours. In the middle of this paragraph, the author has analyzed the same results. Too much repetition.
3. I suggest that the language can be improved by a fluent English speaker or an English editing service.

Minor:
1. Some of the references are very old and lack timeliness. It is best to use the literature of the last ten years.
2. In keywords, the expression pattern should be changed to expression patterns.
3. In the discussion, the preface of the author's description is too long.

Experimental design

which part of the leaves was harvested for RNA extraction?

Validity of the findings

no comment.

Reviewer 2 ·

Basic reporting

Please see comments attached bellow

Experimental design

Please see comments attached bellow

Validity of the findings

Please see comments attached bellow

Annotated reviews are not available for download in order to protect the identity of reviewers who chose to remain anonymous.

Reviewer 3 ·

Basic reporting

The study is very basic on identification of gene family and many such reports from other plants on APX genes are available.

Experimental design

The design plan is ok but authors have not shown functional validation of genes. in gene expression analysis, authors have given stress for only 48 hours however some genes were showing increasing trend in expression even at 48 hours so in my view study should have been conducted for 72 or more hours to get a complete idea of gene upregulation.

Validity of the findings

The data source is available however the ID of the data used in not provided.

Additional comments

Authors are mentioning in abstract that "Taken together, the information in this study will provide theoretical bases and candidate genes for resistance breeding of Chinese hemp", however, they have studied only APX gene family. They should have done interactome analysis of APX with other genes and do the realtime analysis.
How this will help in resistance breeding is also not very clear.
So these modifications may be done in manuscript for more clarity

---

## Round 0.2 · Minor Revisions

Thank you for your patience and collaborative attitude toward the reviewer's suggestions. Your article is coming to finish for its acceptance. However, a few things should be done. Please carefully read the reviewers' suggestions about your article. If you do not accept one or more of their suggestions, give your reasons.

Reviewer 1 ·

Basic reporting

no comment

Experimental design

no comment

Validity of the findings

no comment

Additional comments

After carefully reading, I think that the manuscript has been improved a lot.

Reviewer 2 ·

Basic reporting

please see attached letter

Experimental design

please see attached letter

Validity of the findings

please see attached letter

Additional comments

please see attached letter

Annotated reviews are not available for download in order to protect the identity of reviewers who chose to remain anonymous.

Reviewer 3 ·

Basic reporting

The authors have revised the manuscript and may be accepted for publication.

Experimental design

Good

Validity of the findings

the finding is supported by lab data.

Additional comments

no comments

---

## Round 0.3 · accepted · Accept

I would like to express my appreciation for your incorporating the referees' suggestions and enhancing your article accordingly. I believe your manuscript is now ready for publication. We look forward to your next article.

Reviewer 2 ·

Basic reporting

please see comments attached

Experimental design

please see comments attached

Validity of the findings

please see comments attached